# ERCC1 Overexpression Increases Radioresistance in Colorectal Cancer Cells

**DOI:** 10.3390/cancers14194798

**Published:** 2022-09-30

**Authors:** Yi-Jung Huang, Ming-Yii Huang, Tian-Lu Cheng, Shih-Hsun Kuo, Chien-Chih Ke, Yi-Ting Chen, Yuan-Chin Hsieh, Jaw-Yuan Wang, Chiu-Min Cheng, Chih-Hung Chuang

**Affiliations:** 1Graduate Institute of Medicine, College of Medicine, Kaohsiung Medical University, Kaohsiung 80708, Taiwan; 2Drug Development and Value Creation Research Center, Kaohsiung Medical University, Kaohsiung 80708, Taiwan; 3Department of Radiation Oncology, Kaohsiung Medical University Hospital, Kaohsiung Medical University, Kaohsiung 80708, Taiwan; 4Department of Radiation Oncology, School of Medicine, College of Medicine, Kaohsiung Medical University, Kaohsiung 80708, Taiwan; 5Center for Cancer Research, Kaohsiung Medical University, Kaohsiung 80708, Taiwan; 6Department of Biomedical Science and Environmental Biology, Kaohsiung Medical University, Kaohsiung 80708, Taiwan; 7Department of Medical Imaging and Radiological Sciences, Kaohsiung Medical University, Kaohsiung 80708, Taiwan; 8Department of Medical Research, Kaohsiung Medical University Hospital, Kaohsiung 80708, Taiwan; 9Department of Pathology, Kaohsiung Medical University Hospital, Kaohsiung Medical University, Kaohsiung 80708, Taiwan; 10Department of Pathology, School of Medicine, College of Medicine, Kaohsiung Medical University, Kaohsiung 80708, Taiwan; 11School of Medicine for International Students, I-Shou University, Kaohsiung 84001, Taiwan; 12Division of Colorectal Surgery, Department of Surgery, Kaohsiung Medical University Hospital, Kaohsiung Medical University, Kaohsiung 80708, Taiwan; 13Graduate Institute of Clinical Medicine, College of Medicine, Kaohsiung Medical University, Kaohsiung 80708, Taiwan; 14Department of Surgery, School of Medicine, College of Medicine, Kaohsiung Medical University, Kaohsiung 80708, Taiwan; 15Center for Liquid Biopsy and Cohort Research, Kaohsiung Medical University, Kaohsiung 80708, Taiwan; 16Pingtung Hospital, Ministry of Health and Welfare, Pingtung 90054, Taiwan; 17Department and Graduate Institute of Aquaculture, National Kaohsiung University of Science and Techology, Kaohsiung 81157, Taiwan; 18Department of Medical Laboratory Science and Biotechnology, Kaohsiung Medical University, Kaohsiung 80708, Taiwan

**Keywords:** preoperative concurrent chemoradiotherapy, ERCC1, radiation resistance, HCT116-Tet-on, COLO205-Tet-on

## Abstract

**Simple Summary:**

The 20–30% of locally advanced rectal cancer patients undergoing preoperative concurrent chemoradiotherapy had no expected efficacy, and ERCC1 overexpression was found in these tumor tissue patients. In the interest of confirming and adding details to our understanding of that correlation, The Tet-on gene expression system was used to examine ERCC1 functionality and stability. Our data from regulatable HCT116-Tet-on and COLO205-Tet-on cell lines verified the increased radioresistance in colorectal cancer cells that are associated with ERCC1 overexpression, and they confirmed a high correlation between ERCC1 levels and radiotherapeutic efficiency. Furthermore, overexpression of ERCC1 also increases cell migration under radiation exposure. Additional data from ERCC1 expression regulation in vivo confirmed that the overexpression of increased cancer radiation resistance suggests that ERCC1 expression plays a key role.

**Abstract:**

Preoperative concurrent chemoradiotherapy (CCRT) is a standard treatment for locally advanced rectal cancer patients, but 20–30% do not benefit from the desired therapeutic effects. Previous reports indicate that high levels of ERCC1 reduce the effectiveness of cisplatin-based CCRT; however, it remains unclear as to whether ERCC1 overexpression increases radiation resistance. To clarify the correlation between ERCC1 levels and radiation (RT) resistance, we established two cell lines (HCT116-Tet-on and COLO205-Tet-on), induced them to overexpress ERCC1, detected cell survival following exposure to radiation, established HCT116-Tet-on and COLO205-Tet-on heterotopic cancer animal models, and detected tumor volume following exposure to radiation. We found that ERCC1 overexpression increased radiation resistance. After regulating ERCC1 levels and radiation exposure to verify the correlation, we noted that increased radiation resistance was dependent on ERCC1 upregulation in both cell lines. For further verification, we exposed HCT116-Tet-on and COLO205-Tet-on heterotopic cancer animal models to radiation and observed that ERCC1 overexpression increased colorectal cancer tumor radioresistance in both. Combined, our results suggest that ERCC1 overexpression may serve as a suitable CCRT prognostic marker for colorectal cancer patients.

## 1. Introduction

A total of 104,270 new colon cancer cases, 45,230 new rectal cancer cases, and 52,980 associated deaths were reported in 2021 alone in the United States [1]. Colorectal cancer was identified as the third most prevalent cancer type in the US for that year [2], with local recurrence rates of resected patients with stage II-III rectal cancer ranging from 15% to 65%. Local recurrence rates for stage III patients following total mesorectal excision (TME) surgery have been reported as ranging from 20% to 30% [3,4,5]. Limiting postoperative recurrence is a major treatment priority for colorectal cancer patients.

To improve cure rates and reduce local recurrence, the National Comprehensive Cancer Network recommends preoperative concurrent chemoradiotherapy (CCRT) as a priority treatment for locally advanced rectal cancers [6,7]. Among patients receiving preoperative radiotherapy, one research team has reported a 61% reduction in relative local recurrence risk (95% CI: 0.27–0.58, *p* < 0.0001), with a 6.2% absolute difference at three years (95% CI: 5.3–7.1) and a 24% relative improvement in disease-free survival (HR = 0.76, 95% CI: 0.62–0.94, *p* = 0.013) [8]. According to this study, among others, CCRT treatment of colorectal cancer is now considered indispensable; however, some patients exhibit a poor response to CCRT, thus increasing the potential for tumor progression and additional resection surgery. Three specific issues that must be addressed are (i) patients who do not respond well to CCRT often cannot obtain accurate preoperative pathologic staging for early rectal cancer (T1T2N0), thus increasing the potential for staging overestimates during preoperative clinical imaging, which makes treatment modelling decisions more difficult [9,10]; (ii) additional surgery entails increased difficulties, especially for short-term hypofractionated radiotherapy—more specifically, surgeons must deal with inflamed tumors and the surrounding tissue being represented as congestion or edema, both of which suggest intraoperative bleeding [11]; and (iii) after long-term preoperative radiotherapy, tissue becomes fibrotic, and tumors adhere to surrounding structures, thus making surgical separation more difficult and triggering postoperative complications such as anastomotic leakage (LAR surgery) and perineal wound dehiscence (APR surgery) [3].

Finally, and most importantly, patients who are insensitive to radiotherapy, chemotherapy, and neoadjuvant chemoradiotherapy may experience delayed opportunities for resection surgery, thus allowing for greater tumor development [12]. In short, the tumor edge blur ring in CCRT-insensitive patients stands as a major challenge against successful resection surgery due to inaccuracies in measuring CCRT efficacy potential [11].

We previously reported finding a high correlation between ERCC1 gene overexpression and radiation response in colorectal cancer patients [13]. In the interest of confirming and adding details to our understanding of that correlation, we used controlled gene expression tetracycline technology (Tet-on system) to establish stable ERCC1 expression in two colorectal cell lines to verify the effects of different ERCC1 concentrations on radiation resistance. We then used an ectopic colorectal cancer animal model to investigate the effects, if any, of ERCC1 overexpression on colorectal cancer radiation sensitivity.

## 2. Materials and Methods

### 2.1. Plasmid Construction

The Tet-on ERCC1 system was generated by using a Pas4.1w. Ppuro-aOn cassette (RNAi coil, Academia Sinica, Taipei, Taiwan) and inserting the nucleotide sequence of the ERCC1 isoform1 (also known as: 202, NCBI identifier: P07992-1) which was constructed by PCR amplification using the following primers: CTAGCTAGCAAGCTTGCCACCATGGACCCTGGGAAGGAC, which has a NheI site, and CCGGCGCGCCGTTTAAACATCGATTCACAGATCCTCTTCTGAGATGAGTTTTTGTTCGGG, which has myc Tag and PmeI site.

### 2.2. Cell Lines and Generation of Tet-on Ercc1 Stable Cell Lines

HCT116 and COLO 205 colon cancer cell lines (American Type Culture Collection) were cultured in DMEM (Sigma-Aldrich, St. Louis, MI, USA), supplemented with 10% bovine calf serum, 100 units/mL penicillin, and 100 μg/mL streptomycin at 37 °C in an atmosphere of 5% CO_2_. Tet-on ERCC1 stable cell lines were established via the transfection of a recombinant lentivirus that consisted of a package plasmid, pCMVΔR8.91 (RNAi coil, Academia Sinica, Taipei, Taiwan), pMD.G (RNAi coil, Academia Sinica, Taiwan), and a target plasmid, the Tet-On ERCC1 system, respectively. After selection by 1, 3, and 5 μg/mL Puromycin dihydrochloride (P8833, Sigma-Aldrich), the selection medium was replaced every 2 days for one week.

### 2.3. qPCR and Western Blotting

mRNA extracts were converted to cDNA using a High-Capacity cDNA Reverse Transcription Kit (Biosystems™, Not. 4368814, Thermo Fisher, Waltham, MA, USA). qPCR was followed separately by using a ERCC1 primer protocol (ERCC1 Human qPCR Primer Pair, NM_202001), OriGene Technologies, Inc., Rockville, MD, USA) and a beta Actin (beta Actin Human qPCR Primer Pair, NM_001101, OriGene Technologies, Inc.) primer protocol. Protein extracts, separated by SDS-PAGE and transferred onto NC membranes (Schleicher & Schuell, Keene, NH, USA), were probed with antibodies against ERCC1 (ab 129267, 1:1000, Abcam, Cambridge, UK), an anti-beta-actin antibody (mAbcam 8226, 1:1000, Abcam), or an anti-myc tag antibody [9E10] (ab32, 1:1000, Abcam). Proteins of interest were detected using peroxidase-conjugated AffiniPure Goat anti-Mouse IgG, Fc Fragment Specific (115-035008, 1:1000, Jackson ImmunoResearch, West Grove, PA, USA) or goat anti-rabbit IgG, F(ab’) 2-HRP (sc-3837, Santa Cruz Biotechnology, Dallas, TX, USA) antibodies, and visualized with the Immobilon Western Chemiluminescent HRP Substrate (WBKLS0500, Merck, Kenilworth, NJ, USA), according to the protocol provided. Quantitation of bands was conducted using Image J.

### 2.4. Irradiation Exposes, CELL viability Assay, and Colony Formation Assay

Cells were plated in a 96-well plate (2000/well) and irradiated overnight with 2 Gy per time linear accelerator (Varian Clinac iX, Department of Radiation Oncology, Kaohsiung Medical University Hospital), before a bolus (1.5-cm-thick) was placed on the top and bottom of cell culture plates. Cell viability was measured using an ATPlite kit (510-17281, Blossom Biotechnologies, Taipei, Taiwan), and the luminesce value was measured using a multimode plate reader (VICTORTM X2, PerkinElmer, Waltham, MA, USA). In the colony formation assay, 2000 cells per well were seeded on a 6-well plate (2000/well) and incubated in an incubator for 14 days after irradiation, using the radiation protocol described above. Using a 1% crystal violet (C0775-25G, MERCK)/methanol (67-56-1, MERCK) buffer, the cells were stained for 10 min, washed with ddH_2_O, dried, and the colonies were counted.

### 2.5. Wound Healing Assay

Cells were seeded in 6-well plates at 80% of confluence in a growth medium. A 0.4-mm wide wound was performed after cells were attached to the plate using a 10 μL tip. Wound images were taken during 48 h using Olympus microscope (Olympus Microsystems). Images were quantified with ImageJ software (image J (ImageJ bundled with 64-bit Java 1.8.0_172, Wayne Rasband, Bethesda, Rockville, MD, USA, https://imagej.net/imagej-wiki-static/Downloads (accessed on 28 June 2022)). The cell-free area was normalized on day 0. The quantification of the cell-free area (%) is as follows: cell-free area (%) = (cell-free area of Day 0, Day 1, or Day 2, mm^3^)/(the cell-free area of Day 0, mm^3^) × 100%.

### 2.6. Xenograft Animal Model and Radiation Exposure Schedule

Specific pathogen-free male BALB/c nude mice were obtained from the National Laboratory Animal Center, Taipei, Taiwan. All animal experiments were conducted in accordance with institutional guidelines and approved by the Animal Care and Use Committee of the Kaohsiung Medical University. Tet-on ERCC1, HCT116, and Tet-on ERCC1 COLO 205 cells were subcutaneously injected into 2 × 10^6^ cells in nude mice, respectively. When tumors reached a size of approximately 50 mm^2^, mice were divided into three groups: buffer alone (control), and doxycycline at 10 or 50 mg/kg/mouse by intraperitoneal injection. Radiation was performed with multiple doses of 2 Gy per fraction (cumulative radiation dose was 6 Gy). There were four mice in each group. After sacrificing the mice, the tumor mass was removed and analyzed for ERCC1 expression. In the in vivo experiment, HCT116 and COLO 205 tumor-bearing mice underwent irradiation. Mice were divided into four groups: Group 1 (-ERCC1-RT), in which mice were not treated with either a doxycycline injection or radiation therapy; Group 2 (+ERCC1-RT), in which mice were injected with doxycycline, but received no radiation; Group 3 (-ERCC1+RT), in which mice were not injected with doxycycline, but were treated with radiation; and Group 4 (+ERCC1+RT), in which mice were injected with doxycycline and received radiation. The tumor size was measured by using digital calipers (VWR International, Radnor, PA, USA), and the tumor volume was determined using the following formula: tumor volume [mm^3^] = (length [mm]) × (width [mm])^2^ × 0.52. Tumor weight was measured using an electronic scale.

### 2.7. Statistical Analysis

Statistical significance was calculated using GraphPad Prism 6.0 with Student’s *t*-test and multiple *t*-tests. Data were considered statistically different at *p* < 0.05.

## 3. Results

### 3.1. Establishing a Doxycycline-Inducible Gene Expression System to Regulate ERCC1 and in HCT116 and COLO205

To select colorectal cancer cells with low ERCC1 expression, we analyzed the expression of ERCC1 mRNA and the protein in HEK293, HCT116, SW620, COLO205, and SW480 cell lines (Appendix A). The results showed the expression of ERCC1 in HCT116 and COLO205 to be lower than other colorectal cancer cell lines; therefore, we used HCT116 and COLO205 for the establishment of Tet on ERCC1 stable cell lines. A Tet-on ercc1-myc Tag system (Figure 1A) was used to construct HCT116 and COLO205 colorectal cell lines to examine ERCC1 expression. We compared untreated cells with cells treated with doxycycline for 24 h to determine the successful establishment of HCT116-Tet-on-ERCC1 and COLO205-Tet-on-ERCC1 cell lines. Both anti-MYC and antiERCC1 antibody detection data indicate that compared with a mock wild-type cell line and the non-doxycycline (−) group, the doxycycline group (+) expressed high levels of ERCC1 in both cell lines (Figure 1B,C)—thus, it is evident that HCT116 and COLO 205 were successfully established.

### 3.2. ERCC1 Overexpression Confers Radiation Resistance in-Tet-on-ERCC1 Cell Lines

Cells were divided into three groups: WT (wild type), a Tet-on group without doxycycline treatment Tet-on-ERCC1 (−), and a Tet-on-ERCC1 (+) group. These groups were treated with 2 μg/mL doxycycline for 24 h. Cell viability was analyzed using an ATPlite kit 72 h after radiation exposure at 6–16 Gy. Half-maximal inhibitory concentrations for the HCT116 cell line were 11.17 ± 0.13 Gy for mock wt cells, 9.75 ± 0.06 Gy for Tet-on cells, and 14.12 ± 0.83 Gy for Tet-on + Dox cells (Figure 2A). The respective concentrations for the COLO 205 cell line were 10.29 ± 0.11 Gy, 10.43 ± 0.25 Gy, and 11.65 ± 0.25 Gy (Figure 2B). These results confirm radiation resistance in HCT116-Tet-on-ERCC1 and COLO205-Tet-on-ERCC1 cells conferred by ERCC1 overexpression. To further confirm that the increased radiation tolerance by overexpression of ERCC1 is due to cell proliferation, we used the colony formation for analysis, and HCT116-Tet-on-ERCC1 and COLO205-Tet-on-ERCC1 cells were treated. Then, 0, 0.5, 1.0, 2 μg/mL doxycycline was given with 8 Gy of radiation, and they were left to react for 14 days; the number of colonies formed in each group was subsequently analyzed (%). The results suggested that the number of cell populations in the overexpressed ERCC1 group is greater than the different radiation doses in the HCT116-Tet-on-ERCC1 (Figure 2C,E) and COLO205-Tet-on-ERCC1 cell groups (Figure 2D,F). This phenomenon indicates that the radioresistance of overexpressed ERCC1 reduces the severity of restrictions placed on radiation therapy when treating cell proliferation.

### 3.3. Enhanced Radiation Resistance Depends on HCT116-Tet-on-ERCC1 and COLO205-Tet-on-ERCC1 Upregulation

To verify the correlation between radiation resistance and changes in ERCC1 levels, HCT116-Tet-on-ERCC1 cells or COLO205-Tet-on-ERCC1 cell lines were treated with 0, 0.5, 1, or 2 μg/mL doxycycline to induce a range of ERCC1 expression levels. Cell lysates were collected after 24 h and examined by Western blotting to detect ERCC1 expression. ERCC1/β-actin ratios were measured as 0, 0.20, 0.25, and 0.40 in HCT116-Tet-on-ERCC1, and the cells were treated with 0, 0.5, 1, and 2 μg/mL doxycycline, respectively. In COLO205-Tet-to-ERCC1 cells, the ratios were also 0.03, 0.2, 0.44, and 1, respectively. Combined, these results indicate that the ERCC1 levels in HCT116-Tet-on-ERCC1 and COLO205-Teton-ERCC1 can be regulated via the precise control of doxycycline concentrations (Figure 3A,B). For further verification, the viability of HCT116-Tet-on-ERCC1 and COLO205-Tet-onERCC1 cells exposed to 8 Gy of radiation were analyzed. Results indicate that there was a 100 ± 5.4%, 45.01 ± 1.59%, 54.58 ± 1.53%, 71.25 ± 0.78%, and 78.30 ± 2.47 cell viability for the HCT116-Tet on-ERCC1 cells treated with 0, 0.5, 1, and 2 μg/mL doxycycline, respectively. For the COLO205-Tet onERCC1 cells, the percentages were 100 ± 2.25%, 48.03 ± 3.88%, 70.57 ± 4.94%, 80.74 ± 1.41%, and 92.07 ± 1.98%, respectively (Figure 3C,D). These observations confirmed that radioresistance was dependent on ERCC1 upregulation levels in both cell lines.

### 3.4. Overexpression of ERCC1 Enhances the HCT116-Tet-on-ERCC1 and COLO205-Tet-on-ERCC1 Migration after Radiation Exposure

To clarify whether the overexpression of ERCC1 and radiation exposure following ERCC1 overexpression could affect the proliferation and migration ability of colorectal cancer cells, HCT116-Tet-on-ERCC1 and COLO205-Tet-on-ERCC1 were divided into no expression ERCC1-, overexpression ERCC1 (treated with 2 μg/mL), no expression ERCC1 combined with radiation exposure, and overexpression ERCC1 combined with radiation exposure; ATP Lite analysis was performed at 24 h and 48 h. The results suggest that compared with the non-ERCC1 expression group, the luminescence intensity of overexpressed ERCC1 is not higher than that of the non-ERCC1 expression group, indicating that the presence or absence of overexpressed ERCC1 has no correlation with colorectal cancer cells. Furthermore, to analyze whether the overexpression of ERCC1 has an effect on cell proliferation under radiation irradiation, the results show that under radiation irradiation, the overexpression of ERCC1 can increase cell proliferation in HCT116 Tet on (Figure 4A) and COLO205 Tet on (Figure 4B). Furthermore, we analyzed whether the overexpression of ERCC1 in this environment could increase the migration ability of HCT116 Tet on and COLO205 Tet on. The results showed that the overexpression of ERCC1 in both HCT116 Tet on (Figure 4C) and COLO205 Tet on (Figure 4D) increased their migration ability when exposed to radiation (Figure 4E,F). This phenomenon indicates that the overexpression of ERCC1 could increase the proliferation and migration ability of colorectal cancer cells in a radiation-irradiated environment. We evaluated the association of the ERCC1 level with/without radiation exposure between the markers of MET and EMT in rectal cancer cells. HCT116-Tet-on-ERCC1 and COLO205-Tet-on-ERCC1 were treated with 0 μg/mL or 2 μg/mL Dox to induce the overexpression of ERCC1. After 24 h, the cells were exposed with/without the 8 Gy radiation. The results suggested that the ERCC1 level, and the level of radiation that the cells were exposed to, were not key in terms of affecting tumor generation (Appendix A). This is because the overexpression of ERCC1, and the exposure to radiation, did not alter the levels of MET and EMT markers in CRC cell lines compared with controls; this is perhaps due to the fact that ERCC1 is a downstream factor of DNA repair and does not affect tumorigenesis. Based on the above data and the aforementioned arguments of this study, the overexpression of ERCC1 may only regulate resistance to chemotherapy or radiotherapy without affecting tumorigenesis.

### 3.5. Establishing an Animal Model for Precise ERCC1 Regulation

HCT116-Tet-on and COLO205-Tet-on heterotopic cancer animal models were established to verify the induction of ERCC1 expression in tumors via in vivo injection of doxycycline. Tumors were subcutaneously injected into mice to generate HCT116-Tet-on and COLO205-Tet-on tumor-bearing mice. Doxycycline (10 mg/kg or 50 mg/kg) was IP injected when the tumor volume reached 50 mm^3^. Tumors were collected 24 h post-injection and examined using Western blotting to detect ERCC1 expression. Results show that doxycycline treatment at 50 mg/kg was capable of regulating ERCC1 overexpression in vivo (Figure 5A,B), thus confirming the successful establishment of an animal model for precise ectopic cancer ERCC1 regulation.

### 3.6. ERCC1 Overexpression Induced Radiation Resistance and Reduced Radiationtherapeutic Efficiency

HCT116-Tet-on or COLO205-Tet-on tumors were subcutaneously injected into nude mice to generate HCT116-Tet-on-ERCC1 or COLO205-Tet-on-ERCC1 tumor-bearing mice. Mice were IP-treated with 50 mg/kg doxycycline or PBS and exposed to 6 Gy radiation in three fractions over 7 days. Tumors were measured every 3 days for 33 days (Figure 6A,B), characterized (Figure 6C,D), and then weighed (Figure 6E,F) at the end of the experiment. Our results indicate that there was a larger tumor volume and higher tumor weight in the +Dox +RT group compared with the +RT group on day 33. These results confirm the increased tumor resistance to radiation due to ERCC1 overexpression.

## 4. Discussion

The Tet-on gene expression system was used to examine ERCC1 functionality and stability. Our data from regulatable HCT116-Tet-on and COLO205-Tet-on cell lines verified the increased radioresistance in colorectal cancer cells that are associated with ERCC1 overexpression, and they confirmed a high correlation between ERCC1 levels and radiotherapeutic efficiency. Additional data from ERCC1 expression regulation in vivo confirmed that the overexpression of increased cancer radiation resistance suggests that ERCC1 expression plays a key role.

We found that ERCC1 levels affected chemotherapy and radiation therapy efficacy via DNA repair. Multiple CCRT-associated resistance factor studies have reported high correlations between ERCC1 expression and CCRT clinical efficacy. Xu et al. reported respective CCRT response rates of 76.9% and 56.6% for ERCC1^−^ and ERCC1^+^ patients with regionally moderate advanced nasopharyngeal carcinomas [14,15], and Jun et al. reported a significantly higher three-year overall survival rate in patients with low ERCC1 expression (91.7; 95% CI, 76.0–100.0%) compared with high ERCC1 expression patients (45.5; 95% CI, 23.9–67.1%; *p* = 0.013). Combined, these data suggest that ERCC1 overexpression leads to CCRT resistance. Other research teams have reported high correlations between low ERCC1 expression and in vitro sensitivity to cisplatin in malignant cervical cancer cell lines [16,17], testicular cancer [18], and malignant effusions collected from non-small cell lung cancer patients [19,20]. One retrospective analysis showed a relationship between increased ERCC1 mRNA or protein expression levels and resistance to cisplatin-based chemotherapy in many types of advanced tumors, including non-small cell lung cancer (NSCLC) [21,22], esophageal cancer [23], and colon cancer [24,25,26,27]. In addition, we analyzed the effect of ERCC1 on colorectal cancer patients in the GEPIA database. According to the GEPIA database analysis, colon adenocarcinoma cell lines showed a higher expression of ercc1 mRNA compared with normal cells. Furthermore, we analyzed the correlation between the expression of ercc1 mRNA with the overall survival rate and disease-free survival rate in patients with colon adenocarcinoma. The results showed that the overexpression of ERCC1 did not affect the overall survival rate and disease-free survival rate in patients with rectal cancer; however, our previous study found that the overexpression of ERCC1 in colorectal cancer patients undergoing concurrent chemoradiotherapy has a poor prognosis [28]. Initially, we considered that the overexpression of ERCC1 would only affect staged outcomes of chemotherapy and radiotherapy in patients with rectal cancer, but not the overall survival rate.

The possible mechanism of the expression of ERCC1 on radiotherapy sensitivity could be due to the fact that radiation could cause double-strand breaks and single-strand breaks in cellular DNA through free radicals to limit the progression of cancer [29,30]. Many studies reported that ERCC1 plays an important role in the repair of DNA double-strand breaks and DNA single-strand breaks [31,32]. In double-strand breaks, the ERCC1/XPF complex would recognize and cleave AT-rich DNA sequences to facilitate subsequent DNA replication or conjugation. Furthermore, ERCC1 would stabilize DNA repair and accelerate the process of DSB repair [33,34]. In the process of single-stranded DNA repair, ERCC1-XPF cuts the damaged strand 5′ to remove the damaged base, creating 3′-OH in order to prime DNA synthesis to complete DNA repair and reduce cancer cell apoptosis [35,36]. Based on these mechanisms, we hypothesized that the overexpression of ERCC1 would increase the ability of DNA to repair, and that it would help increase DNA’s tolerance to radiotherapy.

The Tet-on gene expression system offers real-time and precise regulation of cell cycle, inflammatory, DNA repair, and other factors requiring short-term regulation. To study cell cycle factors, Kok et al. regulated Tet-on Cyclin E1 and Tet-a on Cdc25A expression in RPE1 cells [37]. When addressing DNA repair, Mladenov et al. used the Tet-on system to determine the role of the ATR promoter in the repair process [38]. Note that ERCC1 is a DNA damage-induced protein involved in nucleotide excision repair, DNA interstrand crosslink repair [39], and DNA double-strand break repair [40,41]. Certain homologous recombination repair and non-homologous end-joining pathways depend on the ERCC1-XPF function [42,43,44]. Tet-on is being increasingly used to study DNA repairrelated genes such as ERCC1 due to its precision when regulating expression. Since DNA repair-related proteins are only induced by DNA damage, rather than continuously expressed in cells, Tet-on is believed to provide accurate simulations of natural processes in vivo. According to our results, the precise control of ERCC1 expression was doxycycline concentration-dependent both in vitro (Figure 3A,B) and in vivo (Figure 5A,B), thus confirming the appropriateness of using Tet-on to analyze the ERCC1 system. To clarify whether the overexpression of ERCC1 affects other DNA repair pathways, we analyzed the expression of XPF, ATM, and ATR in ERCC1-overexpressing CRC cells. The results indicate that the overexpression of ERCC1 would only have increased the level of XPF. In this isoform, XPF is a DNA endonuclease, and ERCC1 is an auxiliary protein that maintains the function of XPF (Appendix A); therefore, we may speculate that the overexpression of ERCC1 could increase the stability of XPF and reduce the degradation of XPF.

Biomarkers are commonly used for colorectal cancer diagnosis, prognosis, and treatment purposes. Kirsten rat sarcoma virus (KRAS) and N-BRAF mutations help physicians determine the potential for anti-EGFR therapy in terms of increasing EGFR antibody therapy efficacy [45]. CA 19-9 and carcinoembryonic antigens are known to improve early colorectal cancer detection and recurrence following colorectal cancer surgery [46,47]. Microsatellite instability testing is also widely used for chemotherapy prognostic assessment purposes [48]; however, there are no biomarkers that can be used as CCRT prognostic predictors, even though it is considered an important therapeutic regimen.

Data from clinical staining studies have shown that colorectal cancer patients with a poor prognosis following the combination of FOLFOX and radiation exposure often exhibit high levels of ERCC1 overexpression [49]. To confirm the effects of ERCC1 expression on CCRT efficacy, we induced ERCC1 overexpression in colorectal cancer cells and in an ectopic carcinoma animal model prior to radiation exposure, and we found evidence that ERCC1 overexpression triggered resistance to radiotherapy (Figure 2, Figure 3, Figure 4 and Figure 5). According to recent reports, microsatellite instability can be used as an indicator of FOLFOX clinical efficacy, thus suggesting that DNA repair robustness can affect colorectal cancer responsiveness to chemotherapy and radiotherapy. There is currently interest in testing an ERCC1 inhibitor as a CCRT sensitizer.

## 5. Conclusions

Our in vitro and in vivo results indicate that ERCC1 overexpression triggered colorectal cancer radioresistance; therefore, we believe that ERCC1 levels may affect chemotherapy and radiation therapy efficacy via DNA repair. Further research is required to confirm ERCC1 expression detection as an appropriate predictor of CCRT treatment efficacy, and to test the possibility of combining ERCC1 inhibitors to improve CCRT efficacy for colorectal cancer patients.

## Figures and Tables

**Figure 1 cancers-14-04798-f001:**
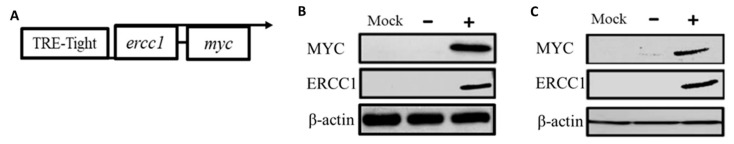
Characterization of Tet-on ERCC1 Colorectal cancer cell lines. (**A**) Schematic diagram of the Tet-on system for assessing doxycycline-inducible ERCC1 expression. (**B**) HCT116 and (**C**) COLO205 cells were categorized as wild type (WT), Tet-on-ERCC1− (no doxycycline treatment) or Tet-on-ERCC1+ (treated with 2 μg/mL doxycycline). Cytoplasmic extrusion mass was collected 24 h post-treatment and used for exogenous ERCC1-MYC tag detection with anti-MYC tag or ERCC1 antibodies.

**Figure 2 cancers-14-04798-f002:**
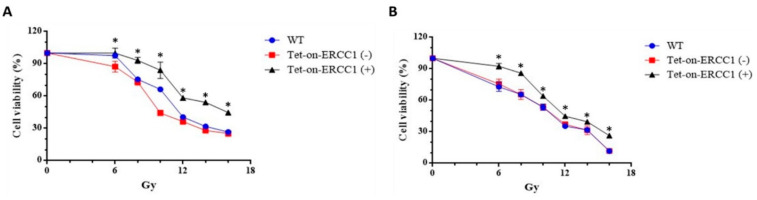
Cell viability was analyzed following ERCC1 overexpression and radiation treatment. Data are shown for (**A**) HCT116 and (**B**) COLO205, including wild type (WT), Tet-onERCC1 cells treated with 2 μg/mL doxycycline, and untreated cells. All relative luminescence unit (RLU) data were collected following radiation treatment for 24 h. Cell viability percentages were calculated as (abs*_sample_* − abs*_blank_*)/(abs*_control_* − abs*_blank_*) × 100. Colony formation assays of (**C**) HCT116 and (**D**) COLO205, including wild type (WT) and Tet-onERCC1 cells (**E**,**F**) were statistically significant (*p* < 0.05 *).

**Figure 3 cancers-14-04798-f003:**
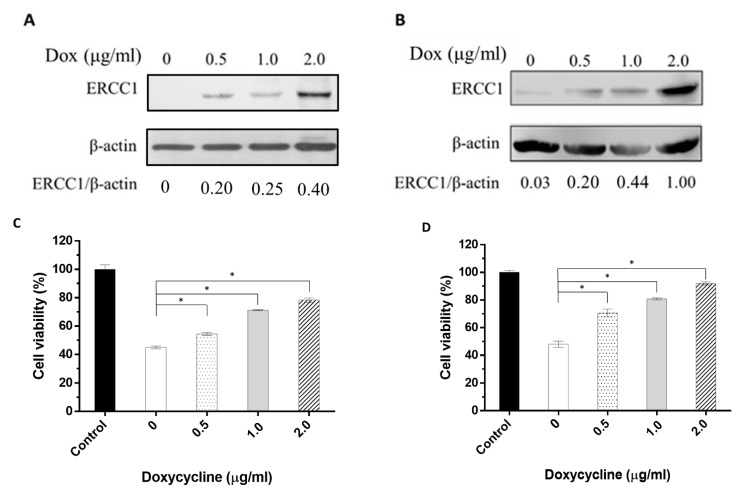
The correlation between the changes in ERCC1 expression level and radiation resistance. The cell extracts were collected from (**A**) HCT116-Tet-on-ERCC1 or (**B**) COLO205-Teton-ERCC1 cells treated with 0, 0.5, 1, or 2 μg/mL doxycycline for 24 h. Anti-ERCC1 or anti-β-actin mAbs were used to detect ERCC1 or β-actin. (**C**) COLO205-Tet-on and (**D**) HCT116-Tet-on cells were treated with 0, 0.5, 1, or 2 μg/mL doxycycline for 24 h following exposure to 8 Gy radiation for 48 h. Cell viability (%) = the luminescence units of the sample groups/the luminescence units of the control group mean × 100%. * indicates a significant difference, *p* < 0.05. The luminescence units were measured by multimode plate reader (VICTORTM X2, PerkinElmer).

**Figure 4 cancers-14-04798-f004:**
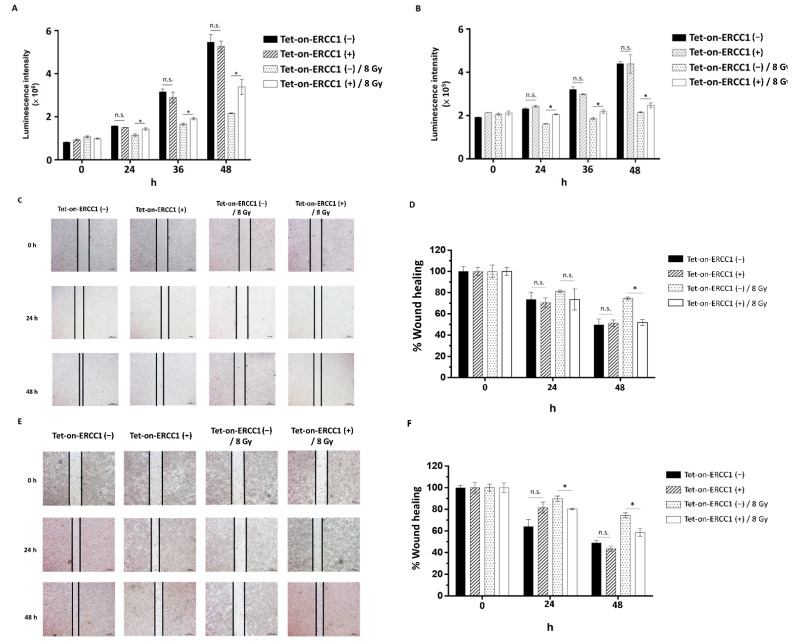
The effect of ERCC1 overexpression on cell proliferation and migration. HCT116-Tet-on-ERCC1 (**A**) and COLO 205-Tet-on-ERCC1 (**B**) were divided into a non-overexpression ERCC1 group, a Tet-on-ERCC1 (−) overexpression ERCC1 group, a Tet-on-ERCC1 (+) combined Tet-on-ERCC1 (−) with radiation exposure group (Tet-on-ERCC1 (−)/8 Gy), and a combined Tet-on-ERCC1 (+) with radiation exposure group (Tet-on-ERCC1 (+)/8 Gy). The luminescence intensity was analyzed at 0 h, 24 h, 36 h, and 48 h. The abovementioned groups were seeded with 2 × 10^5^ into 6 well dishes, and wounds were created at 0 h and photographed at 0 h, 24 h, and 48 h. The percentage of wound healing in HCT116-Tet-on-ERCC1 (**C**,**D**) and COLO205-Tet-on (**E**,**F**). Cell-free area (%) = (cell-free area of Day 0, Day 1, or Day 2, mm^3^)/(the cell-free area of Day 0, mm^3^) × 100%. * indicates a significant difference, *p* < 0.05. n.s. indicates no significant difference.

**Figure 5 cancers-14-04798-f005:**
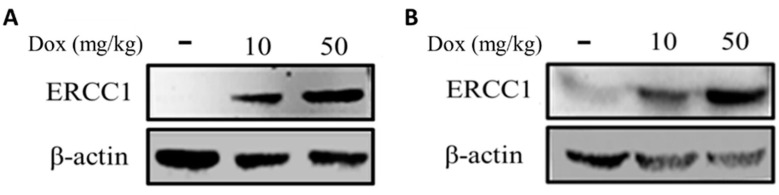
Analysis of ERCC1 levels in Tet-on ERCC1 tumors. ERCC1 expression in (**A**) HCT116-Tet-on and (**B**) COLO205-Tet-on tumors were determined following subcutaneous injections in mice and prior to an IP injection with doxycycline at 10 or 50 mg/kg. Tumors extracts were collected to detect ERCC1 and β-actin levels.

**Figure 6 cancers-14-04798-f006:**
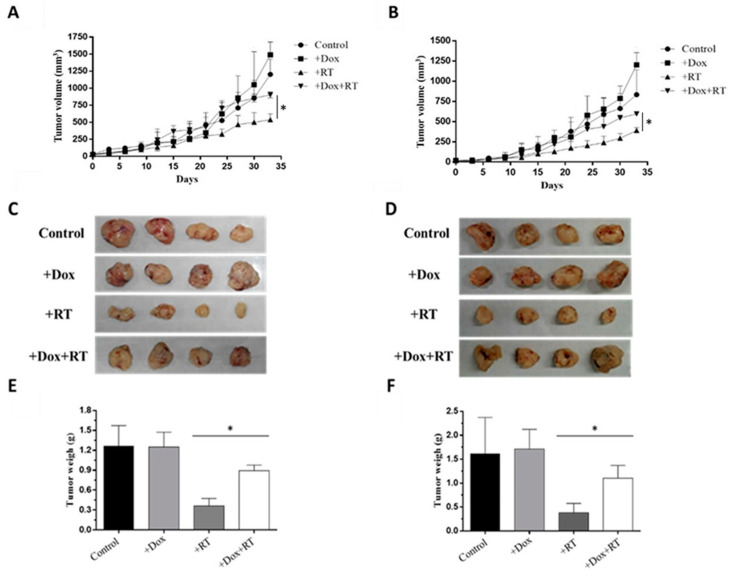
ERCC1 overexpression reduced RT-therapeutic efficiency. A xenograft model was divided into four groups: control, +Dox (treated with doxycycline), +RT (exposed to radiation), and +Dox +RT (treated with doxycycline and exposed to radiation). Tumor volumes were measured between days 0 and 33 in (**A**) HCT116-Tet-on ERCC1 and (**B**) COLO205-Tet-on-ERCC1 cells. (**C**) HCT116 (**D**) and COLO 205 tumors were collected and weighed on day 33. Respective weight data are shown as subfigures (**E**,**F**) (*p* < 0.05 *).

## Data Availability

The data used to support the findings of this study are included in the article.

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
