# Peer review of "ERCC1 Overexpression Increases Radioresistance in Colorectal Cancer Cells"

_cancers, 2022, doi:10.3390/cancers14194798_

Round 1

Reviewer 1 Report

Huang et al. show here that ERCC1 promotes radioresistance in colorectal carcinoma cell lines, highlighting an intrinsic cellular mechanism deployed to overcome radiation-induced cell death. These data are interesting and suggest the potential for developing therapies to counter drug/radiation resistance.

A glaring issue with the manuscript is that the results seem over-summarized in several instances. There is enough room for the authors to expand on their results. Scientific language can also be improved throughout the manuscript. The comments below are expected to significantly improve this work and it is hoped that the authors can further polish the manuscript.

-          Given that ERCC confers radioresistance to colorectal cancer cells, one would expect high or atleast moderate levels of endogenous ERCC in colorectal carcinoma cells/tissue. Yet from Fig. 1 there seems to be no endogenous ERCC in both colorectal carcinoma cell lines. This is strange.

The authors should characterize ERCC expression in a panel of colorectal cancer cell lines by both immunoblotting and RT-PCR and further include data from TCGA showing ERCC expression in normal vs cancer contexts as well as its effect on prognoses (Survival, relapse-free survival and metastases).

-          Fig 3c and 3d are poorly presented and confusing. Under no circumstances can cell viability be greater than 100%. The authors need to repeat the expt and present the data in a sensible/meaningful way.

-          Colony formation assays with cells exposed to increasing doses of radiation would better highlight the resistance of the ERCC-expressing cells to radiation. The authors should include this assay.  

-          One wonders if ERCC promotes other aspects of oncogenesis. Cell proliferation and invasion/migration assays are strongly encouraged to be assessed both with and without ERCC induction and in irradiated vs non-irradiated cells.  

-          Results in Fig 5 seem to indicate that ERCC overexpression itself does not promote tumorigenesis. One wonders whether ERCC may promote invasion. Can the authors also assess markers of EMT in their mice tumor tissue or at least in the Tet-on cell lines?

-          The authors’ findings are based on overexpression experiments. In order to confirm the role of ERCC in promoting radioresistance it is important that the authors identify CRC cell line(s) that natively expresses high ERCC; knockdown/out and then assess sensitization to radiation in vitro and in vivo.  

-       There have been several published studies reporting markers of radioresistance in colorectal and other cancer types. Can the authors perform immunoblotting experiments to confirm enhanced expression of these markers upon modulating ERCC expression and in irradiated cells?  

Author Response

Dear Reviewer:

Many thanks for your valuable comments. We attach our response and revised manuscript in the attached files and the Google Drive link: https://drive.google.com/drive/folders/1ZaNh3woTHSCSdHXX2-hBeCCc6GFp-BUd?usp=sharing

Reviewer 2 Report

The authors detected the relationship between ERCC1 levels and radiation and found that ERCC1 overexpression might serve as a suitable chemoradiotherapy prognostic marker for colorectal cancer. It is interesting. However, I still have several comments regarding the experiments: 

1. In Fig 3c and 3d, the statistical analysis is absent. Furthermore, it seemed that there was no difference between 1.0 Dox treatment group and control in Fig 3C. How to explain the results in Fig 3c when these two group exhibited obviously different expression of ERCC1 in Fig 3a? 

2. From the results in Fig 3, we found that there was some effect of the concentration of Dox on the expression of ERCC1. So what concentration used in Fig.2? 

3. How to select the density of radiation in different assay? 

4. In Fig.5a and 5b, the statistical analysis between RT and Dox+RT is missed. 

5. From Fig 5a and 5b, it seemed that overexpression of ERCC1 promoted tumor growth. What the expression of ERCC1 in colorectal cancer? What is the effect of silencing ERCC1 on radiotherapy sensitivity? 

6.Please discuss the possible mechanism of the expression of ERCC1 on radiotherapy sensitivity.

Author Response

(The authors gave the same response as above.)

Round 2

Reviewer 1 Report

The authors have commendably made the much needed changes to their manuscript by incorporating key experiments. Conclusions are now reasonably supported by some of the extended work performed. 

The authors need to further address some comments regarding the revised figures. Comments (in red fonts) are added in-line to the authors' response and attached herewith.  

The only other pending concern lies with the quality of the language. The authors are strongly advised to seek the services of a professional scientific editor. The entire manuscript requires professional scientific editing.    

Author Response

Many thanks to Reviewer 1 for your valuable comments. Attach our response file.

Reviewer 2 Report

It is OK after the revision.

Author Response

Many thanks to Reviewer 2 for your valuable comments. Attach our response file.
